# Toward Understanding Microbial Ecology to Restore a Degraded Ecosystem

**DOI:** 10.3390/ijerph20054647

**Published:** 2023-03-06

**Authors:** Liyan Song

**Affiliations:** 1School of Resources and Environmental Engineering, Anhui University, Hefei 230039, China; 20023@ahu.edu.cn; Tel.: +86-551-6386-1441; Fax: +86-551-6386-1724; 2Chongqing Institute of Green and Intelligent Technology, Chinese Academy of Sciences, Chongqing 400714, China

**Keywords:** microbial therapeutics, degraded ecosystem, fecal microbiota transplantation, bioaugumentation, microbial ecology

## Abstract

The microbial community plays an important role in maintaining human health, addressing climate change, maintaining environmental quality, etc. High-throughput sequencing leads to the discovery and identification of more microbial community composition and function in diverse ecosystems. Microbiome therapeutics such as fecal microbiota transplantation for human health and bioaugmentation for activated sludge restoration have drawn great attention. However, microbiome therapeutics cannot secure the success of microbiome transplantation. This paper begins with a view on fecal microbiota transplantation and bioaugmentation and is followed by a parallel analysis of these two microbial therapeutic strategies. Accordingly, the microbial ecology mechanisms behind them were discussed. Finally, future research on microbiota transplantation was proposed. Successful application of both microbial therapeutics for human disease and bioremediation for contaminated environments relies on a better understanding of the microbial “entangled bank” and microbial ecology of these environments.

## 1. Microbial Community Research

The boom of sequencing technology coupled with developing bioinformatics is revealing highly diverse and complex microbial communities in natural habitats and the human body [1,2]. To enhance our understanding of the Earth’s many microbial communities, large-scale frameworks such as the Earth Microbiome Project (EMP), the National Microbiome Initiative (NMI), and the Human Microbiome Project (HMP) have been initiated and have made significant progress [3]. Accumulating studies show that microbiota are drivers of host health [4,5,6]. Accordingly, emerging technologies based on the mutualism between the microbiota and host health have been developing. Within these technologies, the transplantation of microbiota into degraded ecosystems such as the patient gut [7], deteriorate soil [8,9,10], and degraded water [11,12] has been drawing the great attention of global researchers. Though microbiota transplantation often results in positive effects, the underlying mechanism of the link between the transplanted species and the indigenous species and the effects of the “mixed” microbiota on host stability are unclear. Particularly, the effectiveness of microbiota transplantation in animal guts is intensely debated [13]. Here, we discuss the mechanistic challenges of the effects of microbiota on host stability and rationalize future research directions.

## 2. Fecal Microbiota Transplantation (FMT)

In research on the human gut microbiome, fecal microbiota transplantation (FMT) has emerged as the most effective treatment for recurrent *Clostridium difficile* infection (CDI) [7]. A recent study by van Nood et al. compared traditional CDI treatment, vancomycin, and FMT treatment for CDI patients, and the results showed that the infusion of donor feces was significantly more effective for the treatment of recurrent CDI than the use of vancomycin [14]. In this case, the feces for transplantation contain a whole microbial community of donors. Another strategy for the treatment of multiply recurrent CDI is to provide patients with cultivable keystone gut microorganisms. For example, Seres Therapeutics is developing SER-109, an agent containing around fifty strains of bacteria that have been proven effective in treating C. difficile for the prevention of multiply recurrent CDI. SER-109 received breakthrough therapy designation from the U.S. Food and Drug Administration (FDA) in 2015. However, an eight-week SER-109 phase two ECOSPOR clinical study in 2016 showed that SER-109 treatment did not achieve the study’s primary endpoint of reducing the relative risk of CDI recurrence. With significant improvement in the composition and interaction of the microbial consortia, SER-109, comprising purified Firmicutes spores, is conducting a US Environmental Protection Agency (EPA) phase three clinical trial and has achieved high rates of sustained clinical response in 2022.

The first written record of fecal transplants could date back to the 4th century in China for treating food poisoning or severe diarrhea [15]. In 1958, fecal transplant was used for the treatment of *Pseudomembranous enterocolitis* [16]. FMT aims to “re-establish the balance of nature” within the intestinal flora to remediate the disruption caused by antibiotic treatment [17]. The mechanism of van Nood’s study, which was mentioned above, is probably the re-establishment of the normal microbiota as a host defense against *C. difficile* [14]. Although there is an absence of large-scale studies using FMT therapy, the underlying mechanisms are probably “re-establishment or restoration of healthy/normal microbial community protection against infections with pathogens (e.g., *C*. *difficile*) from unhealthy/disordered microbial communities containing antibiotic-resistant pathogens developed under the press of antibiotic treatment” [7]. Both FMT and SER-109 are adding functional species to a complex system to improve the ability of the gut microbiota to protect from infection. FMT adds the whole feces microbiome to the gut of patients, while SER-109 adds selected functional species (Figure 1).

## 3. Bioaugmentation

Microbiome therapeutics is similar to bioaugmentation. Adopting the same work principle as microbiome therapeutics, bioaugmentation is also an addition of functionally important species to a complex system to improve the bioremediation ability (Figure 1). Bioaugmentation, however, is one of the most controversial biotechnologies in environmental biotechnology, resulting in many failures [11,12]. “Sludge seeding” is an example of successful bioaugmentation and is often used to start up a new set-up bioreactor in wastewater-activated sludge treatment plants [18]. “Sludge seeding” involves transplanting an entire microbial community, where the community must be taken from a system having a similar working principle as the augmented one. If a wastewater-activated sludge treatment system is unable to breakdown the waste efficiently, isolated sourcing of microbial strains through selective enrichment culture is often used to improve the bioremediation ability [11,12]. These selected strains have defined and essential functions that are used as bioaugmentation agents [19]. However, this approach has had serval failures in improving the bioremediation ability of wastewater-activated sludge treatment [19]. Adversely, “sludge seeding” is the ultimate approach to resolve this problem if selected functional strains or other strategies such as modification of wastewater key physicochemical parameters (e.g., adding a carbon source) and changing the operation mode do not work.

## 4. Microbial Ecology Mechanisms behind FMT and Bioaugmentation

Notably, either FMT or sludge seeding is an “old” microbial therapeutics strategy with hundreds of years of history. In contrast, selecting functional species is a relatively “new” microbial therapeutics strategy that originated in the 20th century. A comparison between these two strategies for microbiome therapeutics and bioaugmentation demonstrated that we incompletely understand the nature of changing and recovering microbial communities. Understanding the effects of the microbiota on host stability requires shifting the focus from individual microbial populations to one that considers the community as a whole. Researching how the microbiota affects the host, subsequently, helps us better study the phenotype, metabolism, and physiology of isolated functional strains. The ecology of the gut microbial community Ecology involves 100 trillion bacteria that coexist by competing with each other for nutrients and space while functioning in a consortial manner to digest complex materials (e.g., polysaccharides) [5]. Given the complexity of gut microbiota, it is difficult to envision and evaluate how the structure and function of this community are maintained, disrupted, and restored. Ecological principles can aid in understanding complex host-microbe interactions and their specific functions.

A microbial community is capable of resisting the invasion of foreign microbes [20], usually by competing for nutrients [21]. For example, microbial strains introduced into the gastrointestinal tracts of mammals can disappear within hours [22,23]. At this point, the introduction of either complex microbial communities or selected strains for CDI treatment or bioaugmented activated sludge is considered an invasion of resident species by the indigenous microbial community. If the introduced microbial community comes from an environment having a similar structure and function as that of the indigenous microbial community, that may be accepted by the indigenous microbial community. The introduced microbial community may have similar interaction signals and feedback networks, which are very similar to those of indigenous microbial communities. For example, Li et al. [24] monitored strain populations in fecal samples from a recent FMT study on metabolic syndrome patients and found that donor and recipient strains coexisted for at least three months, with partial donor strains being replaced by related strains of the same species. The selected (through enrichment) functional microorganisms were usually isolated from the host or inhabited an environment like that of the host [7,11]. During enrichment selection, the link between the isolates and other species of the host is usually broken. When the isolates are introduced into the host, how they connect to the indigenous microbial community may become complicated, especially when the indigenous microbial community has redundant functions [20]. If the introduced species have similar functions as some indigenous microorganisms, the indigenous species are likely to have an adaptive advantage. If the introduced microorganisms have specific functions that are absent from the indigenous microbial community, the indigenous community may resist their invasion. The success of introduction (persistence in the host) is believed to depend on their competition and cooperation with indigenous species [7,11]. However, how the individual microorganisms, populations, and communities interact with the indigenous microbial community and how they influence the stability of the microbial community are currently unknown (Figure 1). The relationship between microbial community diversity, function, and stability is complex. Microbial community diversity can impact microbial functional resistance and resilience. Accordingly, general and/or specific microbial community functions may exhibit different responses to disturbance.

Darwin described an ecosystem as an “entangled bank”: “birds singing on bushes, with various insects flitting about, with worms crawling through the damp earth” [25]. Microbial communities are similarly complex, with many organisms combining in a nonlinear manner to form an integrated network with emergent properties [20,26]. It is well known that many responses of the microbial community to environmental gradients are also nonlinear. Networks have been used for modeling microbial communities as a function of environmental parameters and intra-microbial interaction. An excellent example is a study of microbiome stability by Coyte et al. [26]. They used models to predict ecological stability in the gut microbiome and found that a high diversity of species is likely to coexist stably when the system is dominated by competitive, rather than cooperative interactions.

Currently, we accept that a microbial community maintains relative stability through resilience and resistance to environmental changes (disturbances), which are inherited in macroecology. However, there are differences between microbial ecology and macroecology. For example, modeling microbial ecology is based on the trophic levels (e.g., predator/prey relationships) and shows that the architectural properties of trophic networks influence the relationship between network complexity and stability [27]. Given the vast number of members of a typical microbial community (e.g., 500–1000 species in human guts [5]), the numbers of connections per “node” and the feedback between nodes in microbial ecosystem networks are sharply increased [28]. How to weigh those interactions and understand the mechanisms of regulation on the connection strength between each pair of nodes will be critical for the networks involved in microbial community models. Schrödinger pointed out that an organism survives by continually drawing negative entropy (free energy) from the environment [29]. The related thermodynamics can be simply characterized as organisms absorbing energy from the environment for survival and producing metabolites. This basic concept might help to model how microorganisms interact with each other under disturbance.

## 5. Conclusions and Future Directions

Vast numbers of microorganisms live on the Earth. These vast numbers of microbes drive biogeochemical cycles of elements through complex interactions and metabolic processes. Microbial communities in specific habitats such as extreme environments and manageable engineered systems develop specific metabolic abilities and establish unique interaction networks that respond to system disturbance and stability. Therefore, these specific habitats’ microbiota and microbiome study is very helpful for understanding the link between microbiota and host health. Rhizobacteria and mycorrhiza have been successfully transplanted into the degraded soil and significantly improved plant growth. Rewilding the plant’s microbiome is then a good example of microbiota transplantation. Through competition, syntropy, and predation, diverse microorganisms such as bacterial, archaeal, viral, and eukaryotic communities collaborate on ecosystem ecological processes. Therefore, a comprehensive and complete microbial community in an ecosystem (a microbial “entangled bank”) should be studied. This microbial “entangled bank” can be achieved through a simplified microbial community using a synthetic biology approach. In addition, the fast developments throughout the sequencing and the associated muti-omics study are revealing the microbial community composition and function in various disciplines. The accumulating mega data is approaching the indeed nature of microbial community on this planet.

### 5.1. Extreme Environments Models

Microbial communities in extreme environments could be interesting topics for modeling microbial communities since extreme environments are usually the main determinants in shaping the microbial community structure and function [30]. Accordingly, microorganisms might have specific competition and cooperation strategies to adapt to such an extreme environment [31]. For example, microbiota inhabiting extreme environments such as acid mine drainage, saline lakes, and hot springs have higher relative evolution rates than those living in normal environments such as the surface ocean, fresh water, and soil. Extreme conditions were proposed as the main driver in accelerating the evolution [31]. Furthermore, due to the rapid and extensive development of sequencing technology, more and more extreme environmental (e.g., high pressure, high salinity, radiation, high temperature) microbiota and microbiome have been discovered. These extreme environment modeling efforts help us understand how microbial community structure and function respond to certain conditions and predict the stability of microbial communities under specific conditions.

### 5.2. Manageable Engineered Systems

Manageable engineered systems can be compared to large-scale ecosystems and can be used to elucidate some ecological principles [32]. Shaw et al. [33] supplied engineered microorganisms with specific nitrogen and phosphorus sources, and the engineered microorganisms outcompeted the contaminating strains in the bioreactor. This indicated the importance of the substrate in shaping the species composition, and the underlying mechanisms will highlight the feedback of species to the changing environment. Engineered systems like these can also be used to measure and model the cooperation and competitive relationships among species in an ecosystem. There are many manageable bioengineering systems in environmental engineering, such as membrane bioreactors (MBR), activated sludge systems, and fermentation bioreactors. These bioengineered systems are designed to accomplish specific functions by manipulating the power of the microbial community. Long-term successful operation of these bioengineered systems could provide strong examples for studying the relationship between microbial community stability and disturbance.

### 5.3. Rewilding Plant Microbiomes

The microbial community plays a central role in the nutrients and soil organic matter cycles, thereby impacting soil and plant health [9]. For example, plant growth-promoting rhizobacteria (PGPR) have been founded to reduce soil stress and promote plant growth through phytostimulation, biofertilizers, and biocontrol activities [10]. Another example is that arbuscular mycorrhiza fungi (AMF) can efficiently enhance plant nutrient uptake, immobilization, and translocation of heavy metals [8]. These two examples show that microbiota are the foundation of the microbiota-plant-soil ecosystem. Most importantly, these two cases suggest that the degraded ecosystem could be restored if the transplanted microbes could be integrated into the indigenous habitats. Microbial agriculture and microbial food development might offer a way to enhance sustainable food production and human and planetary health. Most recently, rewilding plant microbiomes have been hypothesized as a potential strategy for improving food production by transplantation of beneficial ancestral microbiomes [6]. Though plant domestication plays important role in the human food supply, this process caused a significant reduction in plant genetic diversity since only human-desired alleles were maintained and spread. Consequently, plant domestication also leads to significant changes in soil microbiota composition and function. For instance, legume domestication (e.g., long-term nitrogen fertilization) results in fewer rhizobacteria in the microbiota-legume-soil ecosystem. As a result, fewer rhizobacteria means that more artificial nitrogen than naturally fixed nitrogen by rhizobacteria is needed for legume growth. Therefore, identification and transplantation of beneficial ancestor microbiota, so-called “rewilding plant microbiomes,” show potential advantages for sustainable food production and environmental stress alleviation [6]. Similar to FMT and bioaugmentation, rewilding plant microbiomes transplant specific microbiota and/or microbiomes with specific traits that are not found or depleted in the degraded system. The mechanism study and application of “rewilding plant microbiomes” is also a good example of microbiota transplantation.

### 5.4. Microbial “Entangled Bank”

Bacterial, archaeal, and eukaryotic communities and viruses co-involving in ecosystem element cycles through a multitude of interactions such as competition, syntrophy, and predation. For example, methane anoxic production is a typical co-work between bacteria and archaea [34]. Polysaccharides such as cellulose are first hydrolyzed into glucose by cellulolytic bacteria. Glucose is then catabolized by fermentative microorganisms into short-chain fatty acids (e.g., acetate), alcohols, hydrogen, and carbon dioxide. Hydrogen and acetate are then consumed by methanogens to produce methane. The syntrophy activities of bacteria and methanogens account for methane production. Therefore, the response and dynamics between both microbiota are the key factors for methane production efficiency. Fungi play an important role in the decomposition of monomers (e.g., cellulose) that are the substrate for other microorganisms in many ecosystems. Eukaryotes have a significant impact on organic pollutant degradation in polluted groundwater by predating bacteria and recycling nutrients [35]. Viruses, particularly bacteriophages/phages (i.e., the viruses that infect prokaryotic organisms, including bacteria and archaea), are responsible for bacterial population and diversity in a top-down manner by infection, due to both the lysogenic and lytic cycles of phage infection that would damage the host [36]. In addition, viruses act as reservoirs for the horizontal exchange of exogenous genes. For instance, viruses can obtain antibiotic resistance genes (ARGs) from bacteria via transduction. Accordingly, the interaction between viruses and bacteria is an important mechanism for antibiotic-polluted ecosystem restoration.

Currently, most cases of transplanting microbiota into degraded ecosystems focus on bacteria, and a few cases use fungi. Considering the microorganism diversity and their interaction linkage in ecological processes and the associated mechanisms such as competition, syntropy, and predation, a comprehensive and complete microbial community in the ecosystem (a microbial “entangled bank”) should be given more attention. Given the vast number of microorganisms in many ecosystems, a simplified microbial community harboring the key functions of specific bacterial, archaeal, and eukaryotic communities and viruses should be established for a better understanding of the stability of the ecosystem. This kind of finding provides a fundamental guide for the success of microbiota transplantation into a degraded ecosystem.

### 5.5. Synthetic Biology

Synthetic biology is an emerging and fast-developing research direction that provides new insights into microbial consortia’s structure and function. Synthetic biology aims to design and create novel artificial pathways and biological systems by combining the disciplines of biology, engineering, informatics, chemistry, and physics. Particularly, synthetic biology can be used to simplify the microbial community in an ecosystem through synthetic microbial consortium engineering. Maintaining the ecosystem’s key functions with minimal microorganism consumption is the core of this synthetic, simplified microbial community, the model of the microbial “entangled bank.” A popular strategy to establish a simplified microbial community is substrate selection. First, substrate and series production are individually used to isolate the functional strains. And then the interaction activity of these selected functional strains is studied and assessed. These strains with better syntrophy characters are selected as the core microbiota for the simplified system. The assemblage with the best function and performance is designed as a synthetic, simplified microbial community. Furthermore, synthetic biology can be used to functionally modify strains through functional gene manipulation by methods such as CRISPR (clustered regularly interspaced short palindromic repeats). The modified functional strains might be transplanted into the degraded ecosystem for restoration based on our understanding of the stability of the microbial community and the disturbance of the strain acceptance ecosystem.

### 5.6. Integration of Multi ‘Omics’ Technologies

Integration of multi-omics technologies (metagenomics, metatranscriptomics, metaproteomics, and metabolomics) [1] and improved cultivation techniques [37] have contributed to a better understanding of the constituents and functions of microbial communities. Though metagenomics, metatranscriptomics, metaproteomics, and metabolomics have been widely used in multiple research directions and the link between the microbial community and DNA, RNA, proteins, and metabolites could be established to some degree, the complex interaction network is still unclear in most cases. Therefore, high-throughput cultivation techniques should be given more effort. If the core microbial strains could be cultivated through high-throughput cultivation techniques, their physiology, and biochemistry would be obtained, which are fundamental information for the microbial community. At the same time, these fast-developing technologies generate vast amounts of data, posing an increased risk to the complexity of data, which was supposed to decrease. How to organize and archive data, share the data, and participate in collaborations between multiple disciplines (e.g., human and environmental) is a big challenge [38]. Therefore, we advocate for the integration of the data from multiple disciplines to answer the ecological principal questions such as microbial community structure and function patterns, how microbial communities respond to disturbance and the relationships between microbial community stability and complexity.

### 5.7. Conclusions

The investigation of microbiota and microbiome dynamics during microbiota transplantation-based restoration of the degraded ecosystem is providing new insights into how the transplanted species coevolve with the indigenous species and how the structure and function of this “mixed” microbial community are restored. Transplantation of microbiota into the degraded ecosystem, such as the patient’s gut, soil, and water, shows ecosystem structure and function restoration to a certain degree. It is not yet clear how the microbiota enhance the treated ecosystem’s succession and stability. More work is needed to elucidate the associated mechanisms using a simplified model, developing microbial ecology principles, and emerging biotechnology. A better understanding of the ecological mechanisms underlying different habitats, from human guts to the ocean, soil, and groundwater to engineered systems, will reveal new and efficient strategies for restoring degraded systems, including the human infectious gut and contaminated environment.

## Figures and Tables

**Figure 1 ijerph-20-04647-f001:**
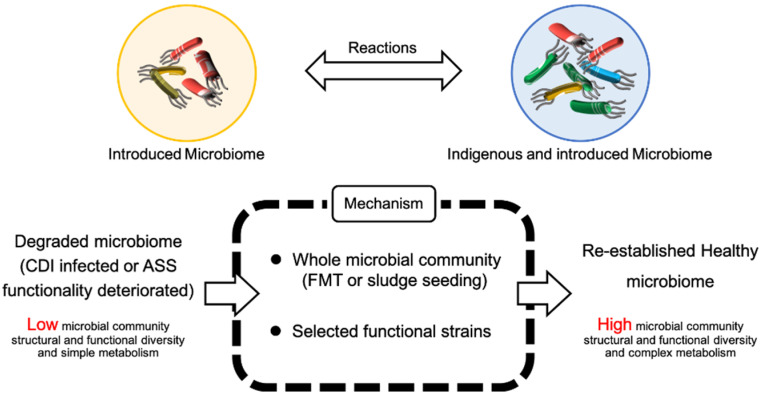
Microbial therapeutics for re-establishment of a healthy microbiome. ASS: activated sludge system.

## Data Availability

Not applicable.

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
