# Peer review of "Toward Understanding Microbial Ecology to Restore a Degraded Ecosystem"

_ijerph, 2023, doi:10.3390/ijerph20054647_

Round 1
Reviewer 1 Report
All my comment are in pdf-file.

Author Response
Response to comments
I would like to thank the reviewer. This was an extremely thorough critique that required considerable effort on the part of the reviewers. The suggestions offered were quite valuable, and have improved the manuscript considerably. I have incorporated the majority of suggestions into the revised manuscript. Below is a detailed response to all comments.
Reviewer 1
Line 21: Capital M
Response: revised per the suggestion.
Line 24 to 27: The second p阿art of the sentence does not have ending. „…and that the balance between the gut microbiota and host health [3, 4].“ What? There should be an answer regarding balance between gut microbiota and host health. Please reformulate the sentence.
Response: The reviewer raises an excellent point. I agreed that the “balance between the gut microbiota and host health” is ambiguous expression. Indeed, I want to express the interactions between gut microbiota and host health. Therefore, I used “mutualism between the gut microbiota and host health” to show the interactions.
Line 34: Clostridium difficile should be in italic. Correct throughout the manuscript.
Response: revised per the suggestion.
Line 48: Explain abbreviation EPA.
Response: revised per the suggestion.
Line 55 to 59: I think it should be written like this: Although there is an absence of large-scale populations for FMT treatment, the underlying mechanism is probably “reestablishment or restoration of healthy/normal microbial community protection against infections with pathogen (e.g., C. difficile) from unhealthy/disordered microbial community containing antibiotic-resistant pathogens developed under the press of antibiotics treatment” [6]
Response: I thank the reviewer for this correction. Accordingly, I revised the sentence per the suggestion.
Line 65: re-establishment of healthy. I do not see what is the purpose of this figure. By my opinion it could be deleted from the manuscript.
Response: I remade this figure to show clear connection between the “transplantation microbiome” and the “indigenous microbiome” that are the key for degraded ecosystem recovery.
Line 68: principle
Response: revised per the suggestion.
Line 69: addition
Response: revised per the suggestion.
Line 83 to 85: Please reformulate the sentence. I do not understand what is „wastewater influence characterization“.
Response: I agreed that “wastewater influence characterization” is ambiguous expression. Accordingly, I revised ““wastewater influence characterization” to wastewater key physicochemical parameters (e.g., addition carbon source).
Line 86: mechanisms
Response: revised per the suggestion.
Line 88: Delete space. You mean „In contrast“?
Response: Thank the reviewer for this correction. I revised the “In contract” to “In contrast”. I also delete the space.
Line 89: 20th century
Response: revised per the suggestion.
Line: 93 to 94: Understanding how the…
Response: I agreed that the sentence is ambiguous expression. Accordingly, I rewrote the sentence. Now the sentence reads:
Understanding the effects of the microbiota on host stability requires shifting the focus from individual microbial populations to one that considers the community as a whole. (Line 103-105)
Line 96: bacteria that coexist
Response: I agreed. Accordingly, I changed the “coexist” to “coexisting”.
Line 119 to 120: Please reformulate the sentence. You mean on redundant functions?
Response: Yes, the reviewer pointed out that the expression of “redundancy functions” should be “redundant functions”. Accordingly, I revised the sentence per the suggestion.
When the isolates are introduced into the host, how they connect to the indigenous microbial community may became complicated, especially when the indigenous microbial community has redundant functions [20].(Line 129-131)
Line 136: environmental
Response: revised per the suggestion.
Line 159: Chapter 5.1. That is correct for extremophiles but human gut or activated sludge could hardly be linked with extreem conditions.
Response: I agreed that extremophiles are extreme condition. Extreme condition may simplify the microbial community structure and function. Therefore, I advocate to use the extreme conditions model to study the associated mechanism.
Line 169: principles
Response: Response: revised per the suggestion.
Line 192: habitats
Response: Response: revised per the suggestion.
By my opinion the last paragraph of the chapter 4 (lines 142 to 157) should be broadened with more ideas/hypotheses from the author and more references. In that way you would add perspective to you paper. Also, human gut and wastewater systems are not the only ecosystems that could benefit from microbial transplantation. As you pointed out in your closing sentence (line 192 to 195) other ecosysems also should be investigated. Why not add in this paper chaper about transplantation of soil microbes from one soil ecosystem to another.
Response: The reviewer raises an excellent point. I agreed. Recently, Rewilding plant microbiomes (Raaijmakers JM, Kiers ET. Rewilding plant microbiomes. Science. 2022 Nov 11;378(6620):599-600) also hypothesized to translate the crop ancestors microbiota to the modern crops for better food productions. The core of the paper is also the “microbiome therapeutics”. Considering the structure of this perspective, we arranged these “Microbiome therapeutics in soil and agriculture” in the introduction and the conclusion part.
In introduction:
Accumulating studies show that microbiota are driver of host health [4, 5, 6]. Accordingly, emerging technologies based on the mutualism between the microbiota and host health have been developing. Within these technologies, transplantation microbiota into degraded ecosystem such as patient gut [7], deteriorate soil [8, 9,10] and water [11, 12] has been drawing great attention of global researchers. Though the microbiota transplantation often results in positive effects, the underlying mechanism of the link of the transplanted species and the indigenous species and the effects of the “mixed” microbiota on host stability is unclear. Particularly, the effectiveness of microbiota transplantation in animal gut is intensely debated [13]. Here, we discuss the mechanistic challenges in the effects of microbiota on host stability and rationalize the future research directions. (Line 31-41)
In conclusion:
5.3 Rewilding plant microbiomes
Microbial community play central role in the nutrients and soil organic matter cycles, thereby impacting the soil and plant health [9]. For example, plant growth-promoting rhizobacteria (PGPR) have been founded to reduce soil stress and promote plant growth through phytostimulation, biofertilizers and biocontrol activities [10]. Another example is that arbuscular mycorrhiza fungi (AMF) can efficiently enhance plant nutrients uptake, immobilization and translocation of heavy metals [8]. These two examples show that microbiota are the foundation of the microbiota-plant-soil ecosystem. Most importantly, these two cases suggest that the degraded ecosystem could be restored if the transplanted microbes could be involvement in the indigenous habitats. Microbial agriculture and microbial food development might offer a way to enhance sustainable food production and human and planetary health. Most recently, rewilding plant microbiomes is hypothesized as potential strategy for improving food production by transplantation beneficial ancestral microbiomes [6]. Though plant domestication plays important roles in food supply for human being, this process caused a significant reduction in plant genetic diversity since only human desired alleles were maintain and spread. Consquently, plant domestication also lead to significant changes in soil microbiota composition and function. For instance, legumes domestication (e.g., long-term nitrogen fertilization) results in the less of rhizobacetria in the microbiota-legume-soil ecosystem. As a result, the less of rhizobacetria means that more artificial nitrogen than rizobacteria naturally fixing nitrogen is needed for legumes growth. Therefore, identification and transplantation of beneficial ancestor microbiota, so-called “rewilding plant microbiomes” show potential advantages for sustainable food production and environmental stress alleviation [6]. Similar to FMT and Bioaugumentation, rewilding plant microbiomes transplants specific microbiota and/or microbiome with specific traits that are not found or depleted in the degraded system. The mechanism study and application of “rewilding plant microbiomes” is also a good example for microbiota transplantation.(Line 218-244)
Beside these revision per the suggestion, the manuscript is required to be enrich per the suggestion of the editor. Accordingly, we added the Microbial “entangled bank” and Synthetic Biology parts for future research on microbiota transplantation. The enriched words were marked purple color.
5.4 Microbial “entangled bank”
Bacterial, archaeal and eukaryotic community and virus are co-involving in ecosystem elements cycles through multitude of interactions such as competition, syntrophy and predation. For example, methane anoxic production is a typical co-work by bacteria and archaea [34]. The polysaccharides such as cellulose is firstly hydrolyzed into glucose by cellulolytic bacteria. Glucose is then catabolized by fermentative microorganism into short-chain fatty acids (e.g., acetate), alcohols, hydrogen and carbon dioxide. Hydrogen and acetate are then consumed by methanogens to produce methane. The syntrophy activities of bacteria and methanogens account for the methane production. Therefore, the response and dynamics between both microbiotas are the key factors for methane production efficiency. In many ecosystem, fungi paly important role in monomers (e.g., cellulose) decomposition that are the substrate for other microorganism. Eukaryotes significant impact organic pollutant degradation in polluted groundwater by predating bacteria and recycling of nutrients [35]. Viruses, particularly bacteriophages/phages (i.e., the viruses that infect prokaryotic organisms, including bacteria and archaea) are responsible for bacterial population and diversity in a top-down manner by infection, due to both lysogenic and lytic cycle of phage infection would cause the host damaged [36]. In addition, virus act as reservoirs for the horizontal exchange of exogenous genes. For instance, virus can obtain antibiotic resistance genes (ARGs) from bacteria via transduction. Accordingly, the interaction between virus and bacteria is the important mechanism for antibiotic-polluted ecosystem restoration.
Currently, most cases in transplantation microbiota into degraded ecosystem are focusing on bacteria and a few cases are using fungi. Considering the microorganism diversity and their interaction linkage in ecological process and the associated mechanism such as competition, syntropy and predation, a comprehensive and complete microbial community in the ecosystem (microbial “entangled bank”) should be paid more attention. Given the vast number of microorganisms in many ecosystems, the simplified microbial community harboring the key functions by specific bacterial, archaeal and eukaryotic community and virus should be established for better understannding the stability of the ecosystem. This kind finding provides fundermental guide for the success of microbiota transplantation into degraded ecosystem.
5.5 Synthetic Biology
Synthetic biology is an emerging and fast-developing research direction that provides new insights for microbial consortia structure and function. Synthetic biology aims to design and create novel artificial pathways and biological systems by combination on the disciplines biology, engineering, informatics, chemistry and physics. Particularly, synthetic biology can be used to simplify the microbial community in ecosystem through synthetic microbial consortium engineering. Maintain the ecosystem key function with minimum microorganism consumption the basic energy and nutrients is the core of this synthetic simplified microbial community, the model of microbial “entangled bank”. A popular strategy to establish simplified microbial community is using substrate selection. First, substrate and the series production are individually used to isolate the functional strains. And then the interaction activity of these selected functional strains is studied and assessed. These strains with better syntrophy characters are selected as the core microbiota for the simplified system. The assemblage with the best function performance is designed as the synthetic simplified microbial community. Furthermore, synthetic biology can be used to functional strain modification through functional genes manipulation by genes edition such as CRISPR (clustered regularly interspaced short palindromic repeats). The modified functional strains might be transplanted into the degraded ecosystem for restoration based on the understanding of the microbial community stability and disturbance of the strain acception ecosystem.
5.6. Conclusion
The investigation of microbiota and microbiome dynamics during microbiota transplantation-based restoration of the degraded ecosystem is providing new insights into how the transplanted species coevolves with the indigenous species and how the structure and function of this “mixed” microbial community is restored. Transplantation microbiota into the degraded ecosystem such as patient gut, soil and water show ecosystem structure and function restoration in certain degree. It is not yet clear that how the microbiota enhance the treated ecosystem succession and stability. More work is needed to elucidate the associated mechanism using simplified model, developing microbial ecology principles and emerging biotechnology. A better understanding of ecological mechanisms underlying different habitats, from human guts, ocean, soil, groundwater, to engineered systems will reveal new and efficient strategies for restoring degraded system, including human infectious gut and contaminated environment.
(Line 246-331)

Reviewer 2 Report
The article (ijerph-2161272 ) “Toward understanding microbial ecology to restore a degraded
ecosystem” is interesting, but before acceptance of this article must be streamlined with following suggestions
Major revision require
1] Section 1: Author must elaborate microbial community and its role in degraded ecosystem, define degraded ecosystem such as soil or water or etc, also add the why restoration is degraded ecosystem require, discuss with very appropriate data and recent citations, please also justified the topic, add suitable hypothesis and proper objectives
2] Section 3 line 68 to 72 must restructure, and correlate with topic of the article see out line for degraded ecosystem and microbes (https://doi.org/10.1007/978-981-16-3840-4_14; https://doi.org/10.1007/s10653-022-01433-3; https://doi.org/10.1155/2022/5275449
3] Line 102 “A microbial community has the ability to resist invasion by foreign microbes “ sthrenth this statement with suitable data and recent updates
4] Line 147 to 149 “Given
the vast number of members of a typical microbial community (e.g., 500-1000 species in
human guts [4]), the numbers of connections per “node” and the feedbacks between nodes 1
in a microbial ecosystem networks are significantly increased” please reframe and add suitable references
5] Conclusion section need to restructure and highlights the significance of the study for future research
6] Author should add attest few tables that justified the findings, and also add one figure related to mechanistic of the study.
Author Response
Response to comments
I would like to thank the reviewers. This was an extremely thorough critique that required considerable effort on the part of the reviewers. The suggestions offered were quite valuable, and have improved the manuscript considerably. I have incorporated the majority of suggestions into our revised manuscript. Below is a detailed response to all comments.
Reviewer 2
Major revision require
1] Section 1: Author must elaborate microbial community and its role in degraded ecosystem, define degraded ecosystem such as soil or water or etc, also add the why restoration is degraded ecosystem require, discuss with very appropriate data and recent citations, please also justified the topic, add suitable hypothesis and proper objectives
Response: The reviewer raises an excellent point. I agreed. Accordingly, I rewrote this part. First, we discussed the microbial community study in natural and human ecosystem and conclude one of the main findings that “microbiota are driver of host health” (Raaijmakers JM, Kiers ET. Rewilding plant microbiomes. Science. 2022 Nov 11;378(6620):599-600). Then, we discussed the hot topic of “transplantation microbiota” into degraded ecosystem for restoration. Meanwhile, the associated gap was concluded. Accordingly, the purpose of this perspective is proposed.
2] Section 3 line 68 to 72 must restructure, and correlate with topic of the article see out line for degraded ecosystem and microbes (https://doi.org/10.1007/978-981-16-3840-4_14; https://doi.org/10.1007/s10653-022-01433-3; https://doi.org/10.1155/2022/5275449)
Response: I thank reviewer for this great suggestion. The “microbiota-plant-soil” study is very interesting and useful. Similar to these mentioned works, rewilding plant microbiomes (Raaijmakers JM, Kiers ET. Rewilding plant microbiomes. Science. 2022 Nov 11;378(6620):599-600) also hypothesized to translate the crop ancestors microbiota to the modern crops for better food productions. Therefore, I used the reviewer mentioned three papers and the “rewilding plant microbiomes” to show the Microbiome therapeutics in soil and agriculture, which is strong support this perspective. Considering the structure of this perspective, we arranged these “Microbiome therapeutics in soil and agriculture” in the introduction and the conclusion part.
In introduction:
Accumulating studies show that microbiota are driver of host health [4, 5, 6]. Accordingly, emerging technologies based on the mutualism between the microbiota and host health have been developing. Within these technologies, transplantation microbiota into degraded ecosystem such as patient gut [7], deteriorate soil [8, 9,10] and water [11, 12] has been drawing great attention of global researchers. Though the microbiota transplantation often results in positive effects, the underlying mechanism of the link of the transplanted species and the indigenous species and the effects of the “mixed” microbiota on host stability is unclear. Particularly, the effectiveness of microbiota transplantation in animal gut is intensely debated [13]. Here, we discuss the mechanistic challenges in the effects of microbiota on host stability and rationalize the future research directions. (Line 31-41)
In conclusion:
5.3 Rewilding plant microbiomes
Microbial community play central role in the nutrients and soil organic matter cycles, thereby impacting the soil and plant health [9]. For example, plant growth-promoting rhizobacteria (PGPR) have been founded to reduce soil stress and promote plant growth through phytostimulation, biofertilizers and biocontrol activities [10]. Another example is that arbuscular mycorrhiza fungi (AMF) can efficiently enhance plant nutrients uptake, immobilization and translocation of heavy metals [8]. These two examples show that microbiota are the foundation of the microbiota-plant-soil ecosystem. Most importantly, these two cases suggest that the degraded ecosystem could be restored if the transplanted microbes could be involvement in the indigenous habitats. Microbial agriculture and microbial food development might offer a way to enhance sustainable food production and human and planetary health. Most recently, rewilding plant microbiomes is hypothesized as potential strategy for improving food production by transplantation beneficial ancestral microbiomes [6]. Though plant domestication plays important roles in food supply for human being, this process caused a significant reduction in plant genetic diversity since only human desired alleles were maintain and spread. Consquently, plant domestication also lead to significant changes in soil microbiota composition and function. For instance, legumes domestication (e.g., long-term nitrogen fertilization) results in the less of rhizobacetria in the microbiota-legume-soil ecosystem. As a result, the less of rhizobacetria means that more artificial nitrogen than rizobacteria naturally fixing nitrogen is needed for legumes growth. Therefore, identification and transplantation of beneficial ancestor microbiota, so-called “rewilding plant microbiomes” show potential advantages for sustainable food production and environmental stress alleviation [6]. Similar to FMT and Bioaugumentation, rewilding plant microbiomes transplants specific microbiota and/or microbiome with specific traits that are not found or depleted in the degraded system. The mechanism study and application of “rewilding plant microbiomes” is also a good example for microbiota transplantation.(Line 218-244)
3] Line 102 “A microbial community has the ability to resist invasion by foreign microbes “ sthrenth this statement with suitable data and recent updates
Response: This is a great suggestion. Accordingly, I added one strong reference (Vila, J. C. C.; Jones, M. L.; Patel, M.; Bell, T.; Rosindell, J., Uncovering the rules of microbial community invasions. Nature Ecology & Evolution 2019, 3, (8), 1162-1171.) to support this statement. Accordingly, I rewrote this sentence. Now the sentence reads:
A microbial community is capable of resisting the invasion by foreign microbes [20], usually by competition on nutrients [21]. (line 109-110)
4] Line 147 to 149 “Given the vast number of members of a typical microbial community (e.g., 500-1000 species in human guts [4]), the numbers of connections per “node” and the feedbacks between nodes in a microbial ecosystem networks are significantly increased” please reframe and add suitable references
Response: This is a great suggestion. Accordingly, I added one strong reference (Hernandez, D. J.; David, A. S.; Menges, E. S.; Searcy, C. A.; Afkhami, M. E., Environmental stress destabilizes microbial networks. The ISME Journal 2021, 15, (6), 1722-1734.) to support this statement.
5] Conclusion section need to restructure and highlights the significance of the study for future research
Response: The reviewer raises an excellent point. I agreed. Accordingly, I added the Microbial “entangled bank” and Synthetic Biology parts for future research on microbiota transplantation. In addition, I also added one conclusion paragraph to strength the objective of this study and the future research directions. The enriched words were marked purple color.
5.4 Microbial “entangled bank”
Bacterial, archaeal and eukaryotic community and virus are co-involving in ecosystem elements cycles through multitude of interactions such as competition, syntrophy and predation. For example, methane anoxic production is a typical co-work by bacteria and archaea [34]. The polysaccharides such as cellulose is firstly hydrolyzed into glucose by cellulolytic bacteria. Glucose is then catabolized by fermentative microorganism into short-chain fatty acids (e.g., acetate), alcohols, hydrogen and carbon dioxide. Hydrogen and acetate are then consumed by methanogens to produce methane. The syntrophy activities of bacteria and methanogens account for the methane production. Therefore, the response and dynamics between both microbiotas are the key factors for methane production efficiency. In many ecosystem, fungi paly important role in monomers (e.g., cellulose) decomposition that are the substrate for other microorganism. Eukaryotes significant impact organic pollutant degradation in polluted groundwater by predating bacteria and recycling of nutrients [35]. Viruses, particularly bacteriophages/phages (i.e., the viruses that infect prokaryotic organisms, including bacteria and archaea) are responsible for bacterial population and diversity in a top-down manner by infection, due to both lysogenic and lytic cycle of phage infection would cause the host damaged [36]. In addition, virus act as reservoirs for the horizontal exchange of exogenous genes. For instance, virus can obtain antibiotic resistance genes (ARGs) from bacteria via transduction. Accordingly, the interaction between virus and bacteria is the important mechanism for antibiotic-polluted ecosystem restoration.
Currently, most cases in transplantation microbiota into degraded ecosystem are focusing on bacteria and a few cases are using fungi. Considering the microorganism diversity and their interaction linkage in ecological process and the associated mechanism such as competition, syntropy and predation, a comprehensive and complete microbial community in the ecosystem (microbial “entangled bank”) should be paid more attention. Given the vast number of microorganisms in many ecosystems, the simplified microbial community harboring the key functions by specific bacterial, archaeal and eukaryotic community and virus should be established for better understannding the stability of the ecosystem. This kind finding provides fundermental guide for the success of microbiota transplantation into degraded ecosystem.
5.5 Synthetic Biology
Synthetic biology is an emerging and fast-developing research direction that provides new insights for microbial consortia structure and function. Synthetic biology aims to design and create novel artificial pathways and biological systems by combination on the disciplines biology, engineering, informatics, chemistry and physics. Particularly, synthetic biology can be used to simplify the microbial community in ecosystem through synthetic microbial consortium engineering. Maintain the ecosystem key function with minimum microorganism consumption the basic energy and nutrients is the core of this synthetic simplified microbial community, the model of microbial “entangled bank”. A popular strategy to establish simplified microbial community is using substrate selection. First, substrate and the series production are individually used to isolate the functional strains. And then the interaction activity of these selected functional strains is studied and assessed. These strains with better syntrophy characters are selected as the core microbiota for the simplified system. The assemblage with the best function performance is designed as the synthetic simplified microbial community. Furthermore, synthetic biology can be used to functional strain modification through functional genes manipulation by genes edition such as CRISPR (clustered regularly interspaced short palindromic repeats). The modified functional strains might be transplanted into the degraded ecosystem for restoration based on the understanding of the microbial community stability and disturbance of the strain acception ecosystem.
5.6. Conclusion
The investigation of microbiota and microbiome dynamics during microbiota transplantation-based restoration of the degraded ecosystem is providing new insights into how the transplanted species coevolves with the indigenous species and how the structure and function of this “mixed” microbial community is restored. Transplantation microbiota into the degraded ecosystem such as patient gut, soil and water show ecosystem structure and function restoration in certain degree. It is not yet clear that how the microbiota enhance the treated ecosystem succession and stability. More work is needed to elucidate the associated mechanism using simplified model, developing microbial ecology principles and emerging biotechnology. A better understanding of ecological mechanisms underlying different habitats, from human guts, ocean, soil, groundwater, to engineered systems will reveal new and efficient strategies for restoring degraded system, including human infectious gut and contaminated environment.
(Line 246-331)
6] Author should add attest few tables that justified the findings, and also add one figure related to mechanistic of the study.
Response: The reviewer pointed out that more table and figure us helpful for this work. I agreed that review paper should provide the up-to-date research information for better understanding on the review topic. This paper focuses on the perspective for the study of microbiome therapeutics that involvement with microbial ecology principles. I compared two typical microbiome therapeutics strategies and then discussed the underlying mechanisms. In addition, I also discussed the microbiome therapeutics in soil and agriculture per the suggestion. Now we believe that this perspective can provide fundamental and mechanistic ideas for readers.
In addition, I remade figure 1 to show clear connection between the “transplantation microbiome” and the “Indigenous microbiome” that are the key for degraded ecosystem recovery.
I really appreciate these professional comments by the reviewer. I am very pleasure to see there are some perspectives (Raaijmakers JM, Kiers ET. Rewilding plant microbiomes. Science. 2022 Nov 11;378(6620):599-600), as well as the works reviewer mentioned that focuses on the microbiome therapeutics in some typical ecosystem. These efforts can significantly contribute on human health and contaminated environment bioremediation.
